

# Community involvement works where enforcement fails: conservation success through community-based management of Amazon river turtle nests

Darren Norris[1,2,3], Fernanda Michalski[2,3,4] and James P. Gibbs[5]

[1] School of Environmental Sciences, Federal University of Amapá, Macapá, Amapá, Brazil
[2] Ecology and Conservation of Amazonian Vertebrates Research Group, Federal University of Amapá, Macapá, Amapá, Brazil
[3] Postgraduate Programme in Tropical Biodiversity, Federal University of Amapá, Macapá, Amapá, Brazil
[4] Instituto Pró-Carnívoros, Atibaia, São Paulo, Brazil
[5] Department of Forest and Environmental Biology, State University of New York (SUNY), Syracuse, NY, United States of America

## ABSTRACT

Law enforcement is widely regarded as a cornerstone to effective natural resource management. Practical guidelines for the optimal use of enforcement measures are lacking particularly in areas protected under sustainable and/or mixed use management regimes and where legal institution are weak. Focusing on the yellow-spotted river turtles (*Podocnemis unifilis*) along 33 km of river that runs between two sustainable–use reserves in the Brazilian Amazon as an illustrative example, we show that two years of patrols to enforce lawful protection regulations had no effect on nest harvesting. In contrast, during one year when community-based management approaches were enacted harvest levels dropped nearly threefold to a rate (26%) that is likely sufficient for river turtle population recovery. Our findings support previous studies that show how community participation, if appropriately implemented, can facilitate effective natural resource management where law enforcement is limited or ineffective.

## INTRODUCTION

Law enforcement can be an important tool for biodiversity conservation (*Hilborn et al., 2006*; *Keane et al., 2008*). Yet without motivation for compliance, punitive governance actions (including enforcement) are unlikely to succeed (*Dietz, Ostrom & Stern, 2003*; *Keane et al., 2008*; *Ostrom, 2015*) and can even be counter-productive by generating conflicts with local communities (*Dietz, Ostrom & Stern, 2003*). The reality is that successful governance through effective external enforcement is the exception not the rule (*Ostrom, 2015*). Further work is urgently required to produce practical guidelines for the optimal use of enforcement measures in biodiversity conservation (*Dietz, Ostrom & Stern, 2003*; *Keane et al., 2008*).

Corresponding author
Darren Norris,
darren.norris@unifap.br

Developing effective conservation solutions is complicated for protected areas managed under multiuse governance regimes (*Lambin et al., 2014*; *Nolte et al., 2013*; *Pfaff et al., 2015*; *Richards et al., 2017*). Rules governing human behavior are at the heart of every system of common pool/multiuse resource management (*Ostrom, 2015*; *Salo, Sirén & Kalliola, 2014*). Within these scenarios governance is the art of motivating stakeholders to follow established rules and is necessary (but not always sufficient) for management success (*Salo, Sirén & Kalliola, 2014*). There are increasing examples where self-regulating community-based management can be equally if not more effective than external enforcement in preventing the over exploitation of natural resources (*Campos-Silva & Peres, 2016*; *Nepstad et al., 2006*). Integrating conservation and development projects can represent a successful, hybridized approach despite governance and enforcement being particularly challenging in mixed-use/common pool resource areas (*Dietz, Ostrom & Stern, 2003*).

Sustainable-use protected areas have rapidly expanded in number and area across the Brazilian Amazon through the 21st century (*Bernard, Penna & Araújo, 2014*; *Peres, 2011*; *Pfaff et al., 2015*). Such areas are a primary example of general attempts to integrate communities and protected areas to generate conservation solutions. Local community-based management can be effective for the conservation of common pool resources (*Campos-Silva & Peres, 2016*) although perennially struggle in practice with the massive spatial scale at which they are often intended to achieve success, physical and intellectual isolation, dearth of funding, and lack of political will (*Peres, 2011*).

River turtles represent an important common pool resource across temperate and tropical regions (*Dudgeon et al., 2006*) and are a focus of local community-based management in the Amazon. Aquatic turtles are one of the most endangered groups of vertebrates (*Gibbon et al., 2000*), with some 52% of river turtles listed in some form of "threatened" by the IUCN (*Böhm et al., 2013*). River turtles represent provisioning (food, source of income) and cultural services for local populations across the globe (*Eisemberg et al., 2011*; *Harju, Sirén & Salo, 2017*; *Mittermeier et al., 2015*). River turtles, therefore, present informative and highly pertinent examples of the challenges facing conservation of common pool resources in a rapidly changing world (*Dietz, Ostrom & Stern, 2003*; *Gibbon et al., 2000*; *Harju, Sirén & Salo, 2017*; *Mittermeier et al., 2015*).

Anthropogenic impacts including overexploitation have led to the decimation of many river turtle populations across the Amazon region (*Castello et al., 2013*; *Mittermeier, 1978*; *Smith, 1979*). Despite recent changes driven by region wide development (*Piperata et al., 2011*) the eggs and meat of side-necked turtles (Podocnemididae) continue to be a widespread component in the diet of both rural and urban peoples across Amazonia (*Harju, Sirén & Salo, 2017*; *Parry, Barlow & Pereira, 2014*). Continued high demand for Amazon river turtles generates high expectations for, and sharp debate about governance-effective management approaches necessary to ensure the conservation of these species (*Páez et al., 2015*).

The most commonly adopted management approaches for Amazon river turtles involve actions around the protection and conservation of nesting areas, nests and hatchlings during this critical life phase when individual turtles are highly concentrated on exposed river sands accessible to people (*Harju, Sirén & Salo, 2017*; *Mittermeier, 1978*;

*Páez et al., 2015*; *Vogt, 2008*). Nesting areas emerge episodically with seasonal lowering of river levels and can shift from one season to the next so are typically common-pool resources for which management actions necessarily include a variety of governance mechanisms to ensure compliance (*Salo, Sirén & Kalliola, 2014*). Mechanisms can be enforcement-focused and punitive or they can target engagement with local stakeholders. The application of enforcement and engagement (independently or in combination), depends heavily on the local context to secure both participation and compliance (*Dietz, Ostrom & Stern, 2003*; *Ostrom, 2015*; *Salo, Sirén & Kalliola, 2014*).

The social, cultural and environmental diversity across Amazonia means that there remain few examples comparing the relative efficacy of different governance approaches in the conservation management of river turtles. How do we know what actually works? What combination of stakeholder engagement, constructive participation, incentivization and external enforcement actually translate into improved population status for river turtles and increased sustainability of resource use for local stakeholders?

Herein we address these questions by presenting a recent harvest history of yellow-spotted river turtle nests along 33 km of river situated between two Amazon sustainable-use protected areas. We compare the exploitation of river turtle nests along the river segment during years when "protection" emphasized only law enforcement via parties external to the local community versus when conservation effort focused only on community engagement for the collaborative management of turtle nesting areas. This "quasi-experiment" in the form of a temporal comparison among years with strongly contrasting management approaches enabled us to evaluate the relative success of community involvement versus enforcement for a sustaining a shared-pool resource in a protected area.

## MATERIALS AND METHODS

### Ethics statement

Ethical approval was not required for our noninvasive study, as we did not collect any biological samples nor interfere with the behavior of the study species. Permission to collect observational data from river turtle nesting areas was provided by research permit number IBAMA/SISBIO 49632-1 and 49632-2 to DN and FM, issued by the Instituto Chico Mendes de Conservação da Biodiversidade (ICMBio). Interviews and meetings with local residents were approved by IBAMA/SISBIO (permits 45034-1, 45034-2, 45034-3) and the Ethics Committee in Research from the Federal University of Amapá (UNIFAP) (CAAE 42064815.5.0000.0003, Permit number 1.013.843).

### Study area

The study was conducted along 33 km of the Falsino River, in the state of Amapá, Brazil (N 0.77327, W 51.58064; Fig. 1). This river segment runs between two sustainable-use protected areas, the Amapá National Forest and the Amapá State Forest (hereafter "FLONA" and "FLOTA" respectively). Both are National Forests, but only the FLONA (VI—"Protected area with sustainable use of natural resources") is designated within the IUCN Protected Area Classification (*UNEP-WCMC & IUCN , 2018*). This particular stretch of river is 61 km

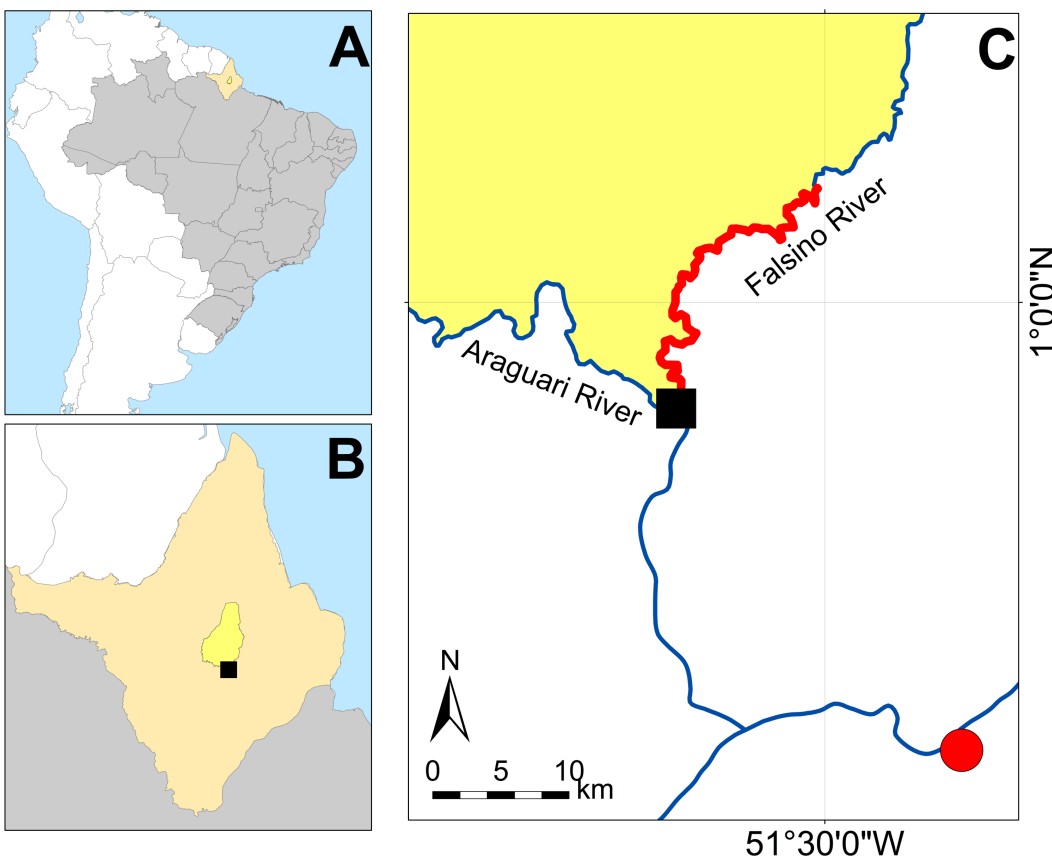

**Figure 1   Study area.** (A) State of Amapá in Brazil. (B) Location within Amapá. (C) Showing location of community managed *Podocnemis unifilis* nesting areas. Red solid line delimits the location of the Falsino river section with community management. Solid black square is the location of the ICMBio base that served as the enforcement base in 2015 and 2016. The nearest town—Porto Grande is shown by a solid red circle. Location of the FLONA sustainable-use protected area is shown in yellow.

from the nearest town and suffers relatively low anthropogenic influence (*De Oliveira, Norris & Michalski, 2015*; *Norris & Michalski, 2013*), hosting just 3–6 houses during our study.

The regional climate is classified by Köppen-Geiger as ''Am'' (Equatorial monsoon) (*Kottek et al., 2006*), with an annual rainfall greater than 2,000 mm (*ANA, 2016*). The driest months are September to November (total monthly rainfall < 150 mm) and the wettest months (total monthly rainfall > 300 mm) from February to April (S1 Fig in *Paredes et al., 2017*).

## Background on river turtle management approaches

The conservation and commercial exploitation of *Podocnemis unifilis* in Brazil contrasts with other South American countries. Conservation actions primarily aimed at protecting nests were initiated at a time of military rule (early 1970's) and have continued through democratization (1970's–1980's) and into the 21st century (*Alho, 1985*; *Páez et al., 2015*; *Vogt, 2008*). Today, management actions are developed within international (*Podocnemis*

*unifilis* is listed on CITES Appendix II, to which Brazil is a contracting party since 1975) and national laws. For example, Brazilian Federal law (Lei 5.197 of 3 January 1967) prohibits the capture of wild turtles. The commercialization of farmed *Podocnemis unifilis* products including meat, eggs and hatchlings is legally (under certain circumstances) and technically possible but these actions are regulated by a complex and ambiguous suit of Brazilian Federal and State laws.

In our study area, the management approaches aimed at conserving the species along the rivers bordering the protected areas have varied over the last decade. Few alternative sources of revenue exist for the local riverine populations in the area particularly as other widely commercialized species such as the Giant South American Turtle (*Podocnemis expansa*) and the arapaima (*Arapaima gigas*) are not present. Interviews with local landowners confirm that these species have not been present for at least the last 60 years. Yet it is not possible to be certain whether this absence comes from historic overexploitation or biogeographic limits to the species distribution. Although surrounded by sustainable-use protected areas the continued survival of the traditional riverine communities is further complicated by mercury contamination of fish stocks (*Venturieri et al., 2017*) and hydroelectric expansion (*Norris, Michalski & Gibbs, 2018*) that degrade the natural resources upon which they depend.

The need for direct conservation action to ensure the survival and recovery of *Podocnemis unifilis* populations comes from anthropogenic pressures (primarily overexploitation e.g., hunting and nest harvesting) originating from both a nearby town (Porto Grande, current population ca. 10,000; *IBGE, 2010*) and the local riverine communities including approximately 50 families that live along the rivers upstream from Porto Grande (*Norris & Michalski, 2013*). The FLONA was created in 1989, and local community members report that the first FLONA manager may have attempted nest translocation and protection activities prior to 2009. Subsequently, in 2012, the organ responsible for the management of the FLONA (ICMBio) implemented nest translocation and protection in an artificial nesting area constructed at the ICMBio base, located at the entrance of the Falsino River (Fig. 1). A small number of community members received payment to participate in the monitoring and protection of these nests. The locals report that this action was of limited success as more than half of the translocated nests did not survive and there was no funding to continue activities in subsequent years.

With the publication of the FLONA management plan in 2014 it became feasible (management plans are a legal pre-requisite for many governance actions associated with Brazilian protected areas) to adopt actions focusing on compliance of existing legislation. With a continued increase in infractions (e.g., illegal hunting) a decision was made by ICMBio managers to initiate external enforcement actions. The enforcement was conducted in 2015 and 2016 by the specialist Environmental Police task force ("Batalhão de Polícia Militar Ambiental"). Funding for enforcement actions came from a collaboration with the neighboring strictly protected (IUCN Category II) Tumucumaque National Park ("Parque Nacional Montanhas do Tumucumaque"). Enforcement was provided to patrol navigable rivers that flow along the borders of the sustainable-use areas (FLONA and FLOTA) and the National Park, i.e., rivers that provide access to the strictly protected National Park.

The National Park receives funding for such activities from the Amazon Region Protected Areas (ARPA) program, whereas the sustainable-use areas do not receive funding for any enforcement actions.

Enforcement patrols included between four to six people and were conducted along more than 160 km of rivers that surround the protected areas, including the 33 km study area. Enforcement patrols focused on checks for illegal activities around the protected areas (such as hunting and the possession of illegal arms) and included stopping boats to check fishing nets, the fish, the boat contents and question boat crews. The enforcement activities also included stops at beaches to check for illegal activities including hunting. During the enforcement period, the police team was based on the ICMBIO base (Fig. 1). One member of the police team was also stationed permanently at the base to monitor and question and/or search any boats that passed this strategic location. Although there was a broad remit for the enforcement patrols (i.e., they were targeting a range of illegal activities) their timing was synchronized with the river turtle nesting season, which was a tactical management decision that aimed to use external enforcement to increase legal compliance and reduce illegal nest harvests and thereby increase the survival of river turtle nests and production of hatchlings along the rivers.

In 2017 a community-based management approach was undertaken, inspired by a request from the local community itself. The decisions as to the actions adopted (e.g., the where, who, what and how) came after two large meetings with the local landowners and ICMBio managers. The community management activities were focused on landowners living along the Falsino River who participated in nest protection activities (predator exclusion devices were placed on top of the river turtle nests to avoid natural predation Fig. S1). Activities were focused around strategic nesting areas that were the larger nesting areas (>4 m$^2$) hosting most of the nests in the 33 km study area (accounting for 90 % of nests in 2011). During the nesting season, local landowners monitored the nesting areas twice a week, taking note, and protecting any new turtle nests with predator exclusion devices (Fig. S1). Local landowners did not receive any payment for participation, but received training, gasoline and materials necessary from an ongoing research project (http://sites.nationalacademies.org/PGA/PEER/PEERscience/PGA_168063). Researchers contacted the landowners every two weeks to receive updates of the nest monitoring.

## Nesting area surveys

We assessed nesting success during four nesting seasons (2011, 2015, 2016 and 2017) along the same 33 km river segment (Table 1). This sequence of monitoring seasons represented a temporally structured quasi-experiment that included one reference season with no enforcement and no community management (2011), two in which external enforcement of existing protection regulations was undertaken (2015, 2016) and one in which a community-based nest protection program was enacted (2017). These temporal differences in management actions along the same river segment enabled us to contrast the relative success of external enforcement and community-based management in protecting river turtle nests.
**Table 1 Nest harvest along the Falsino River.** Harvest of yellow-spotted river turtle (*Podocnemis unifilis*) nests during four years along 33 km of the Falsino River. Estimates of survey and enforcement effort in years with different management approaches also included.

| Year | CBM[a] | Enforce[b] | Total areas surveyed[c] | Total nesting areas (harvested, unharvested[c]) | Total nests (harvested, unharvested) | Nests per area | Nest density[c,d] |
|------|--------|------------|--------------------------|-------------------------------------------------|--------------------------------------|----------------|--------------------|
| 2011 | No | No | | 22 | 161 (121, 40)[e] | 7.3 | |
| 2015 | No | Yes (155) | 87 | 11 (9, 2)[e] | 38 (21, 17)[f] | 3.5 | 83.9 |
| 2016 | No | Yes (355) | 79 | 27 (19, 8)[e] | 69 (46, 23)[e,f] | 2.6 | 50.7 |
| 2017 | Yes | No | 83 | 26 (11, 15) | 144 (38, 106) | 5.5 | 105.9 |

**Notes.**
[a] If community-based management was applied.
[b] If external enforcement patrols were used, with effort (liters of petrol used) during the nesting season in parentheses.
[c] Values not measured in 2011.
[d] Nests per hectare of nesting areas.
[e,f] Denote years with the same proportion of harvest within columns. Pairwise comparisons between years obtained using a 2-sample test for equality of proportions with continuity correction ($P_\alpha = 0.1$).

To quantify levels of nest harvesting a series of nesting area surveys (*Norris, Michalski & Gibbs, 2018*) were conducted between September and December in all study years. These months correspond to low water and include the complete nesting and first half of the hatching season in the study area (D Norris, pers. obs., 2016). Nesting data from 2011 were obtained from a previous study (*Arraes, 2012*). In 2015, 2016, and 2017 we then repeated the methodologies applied in 2011, as briefly summarized here, with full details available in *Norris, Michalski & Gibbs (2018)*. To locate river turtle nests we conducted monthly (interval of 20–30 days between visits) surveys of all potential nesting areas including river banks and islands along the 33 km section by navigating along the river in a motorized boat at a constant speed (ca. 10 km/h).

When potential nesting areas were identified through visually searching river banks and circling islands we stopped to search for turtle nests. We identified potential areas where environmental conditions matched those described in the literature (*Escalona & Fa, 1998*; *Norris, Michalski & Gibbs, 2018*; *Pignati et al., 2013*) and/or those found at the nesting areas from 2011. Nesting areas were mapped with a handheld GPS to calculate the size of the available nesting area (*Escalona & Fa, 1998*; *Norris, Michalski & Gibbs, 2018*). These searches were conducted together with local residents with decades of knowledge of nesting areas and took place independently of any enforcement or community-based management activities. To minimize possible observer biases related to the searches of turtle nesting areas and nests we maintained at least one observer in the team constant while conducting searches in all years (2011, 2015, 2016, and 2017). Naturally depredated nests were identified by the presence of broken eggshells and/or remains of partially eaten eggs outside the nest, disturbed/uncovered nests surrounded by animal tracks and the presence of wildlife excavation marks. Human removal was identified when nests were found open (without sand cover), with a mean depth used by the river turtles (~10–15 cm), but without

eggs or partially eaten eggshells. Human removal of eggs was also usually associated with signs of human activities, such as footprints, fire, charcoal, and campsite on the nesting areas.

## Data analysis

We used the proportion of nesting areas and proportion of nests harvested by humans as response variables to compare the effects of external enforcement and community involvement. The contrast (2-sample test for equality of proportions with continuity correction) in nest harvest proportions between years 2011, 2015 and 2016 enabled us to test the hypothesis that enforcement generated differences in harvest rates. To test the hypothesis that increased enforcement was associated with reduced harvest levels, we examined Spearman correlation between the amount of boat fuel (liters of petrol) used by the enforcement patrols as our index of enforcement effort and the proportion of nesting areas and nests harvested. If enforcement was effective for the management of the river turtles, we predict that the harvest (proportion of both nesting areas and nests harvested) should decline with increasing enforcement effort (represented by liters of petrol used). The contrast (2-sample test for equality of proportions with continuity correction) in nest status between years 2015, 2016 and 2017 enabled us to test the hypothesis that community-based management resulted in lower harvest rates compared with enforcement. All descriptive analysis and graphics production were undertaken within the R language and environment for statistical computing (*R Core Team, 2017*; *Wickham, 2009*).

## RESULTS

Harvest rates were high in enforcement years (averaging 76% and 61% for area and nest harvest proportions respectively), whereas the lowest harvest level occurred when no enforcement patrols occurred along the river (42% and 26% harvest of areas and nests respectively, Table 1, Fig. 2). The nest harvest rate declined during years of law enforcement compared to the 2011 reference level (Table 1, Fig. 2); although nest harvest levels in one enforcement year (2015) did differ compared with 2011 (2-sample test for equality of proportions, $P = 0.02508$), the continually high harvest rates (55% and 67%, 2015 and 2016 respectively, Fig. 2) imply this statistical difference was of little biological relevance. There was no association between enforcement effort and harvest rates over the four years (Spearman rho = 0.11, $P = 0.895$ and Spearman rho = 0.50, $P = 1.00$ for nest and area harvests respectively). We estimated that a twofold increase in enforcement patrol effort (155 to 355 liters) between 2015 and 2016 had no significant effect on area or nest harvest rates (Table 1, 2-sample test for equality of proportions $P = 0.3381$ and $0.7485$ for nest and area harvest respectively). This substantially increased patrolling effort was associated with only a small reduction in the proportion of areas harvested (82% to 70%), yet also a 33% increase in nest harvest levels (55% to 67%, Table 1, Fig. 2).

Notably, the lowest nest harvest was recorded in 2017 when there was no enforcement but community management was implemented (Table 1, Fig. 2). Prior to community-based management, the majority of nesting areas and nests were harvested (Table 1, Fig. 2).

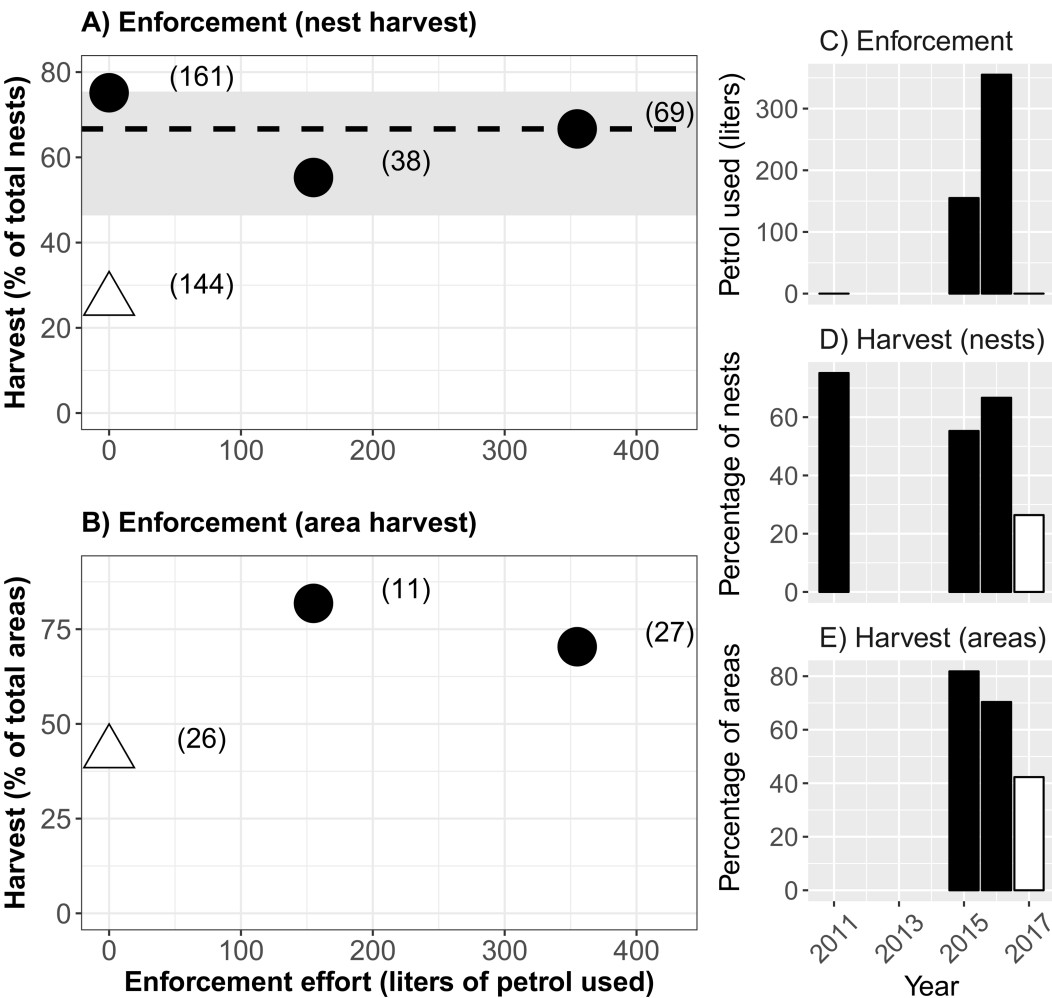

**Figure 2** **Four years of nest harvest.** Harvest of yellow-spotted river turtle (*Podocnemis unifilis*) nests along 33 km of river in Amapá State, Brazil. (A) Nest harvest during years with (white triangle) and without (black circles) community-based management (CBM). Total number of nests in parentheses, the dashed horizontal line shows the median harvest from the three years without CBM. Horizontal grey shading represents the 95% confidence interval of harvest levels in the absence of CBM across the species range (see Table 2). (B) Nesting areas with harvest during years with (white triangle) and without (black circles) CBM. Total number of areas in parentheses, the number of nests harvested per nesting area was not recorded in 2011. It was not possible to estimate confidence intervals for area harvest due to lack of reported results (see Table 2). (C) Enforcement effort during four river turtle nesting seasons. (D) Proportion of river turtle nests harvested in four nesting seasons. (E) Proportion of river turtle nesting areas harvested in four nesting seasons.

This pattern was inverted with community-based management as the majority of nesting areas and nests remained unharvested in 2017 (Table 1, Fig. 2). The community-based management harvest rate in 2017 (42% and 26% harvest of areas and nests respectively) differed significantly from the years (2015 and 2016) with enforcement (Table 1, 2-sample test for equality of proportions, $P < 0.0001$ for both areas and nest harvest proportions). The portion of nests harvested under community-based management (26%) also declined

**Table 2  Nest harvest rates obtained from the literature.** Comparison of harvest rates of yellow-spotted river turtle (*Podocnemis unifilis*) nests with and without community-based management. Means and confidence limits obtained via nonparametric bootstrap without assuming normality.

| Type | Study duration | Nesting areas[a] | | | Nests | | | Location | Source |
|---|---|---|---|---|---|---|---|---|---|
| | | Harvest (%) | Total | Harvest | Harvest (%) | Total | Harvest | | |
| No community management | Single season | – | – | – | 51.9 | 952 | 494 | Manú, Peru | *Landeo (1997)* |
| | Single season | 71.4 | 7 | 5 | 84.9 | 351 | 298 | Nichare-Tawadu, Venezuela | *Escalona & Fa (1998)* |
| | Single season | – | – | – | 50.9 | 165 | 84 | Manapire & Cojedes, Venezuela. | *Hernández et al. (2010)* |
| | Single season | – | – | – | 81.2 | 69 | 56 | Bajo & Medio Putumayo, Peru | *Bermúdez-Romero et al. (2010)* |
| | Multi season | – | 5 | – | 31.8 | 434 | 138 | Iténez & Paraguá, Bolivia | *Lipman (2008)* |
| | Multi season | 76.1 | 19 | 15 | 65.7 | 268 | 188 | Falsino River, Brazil | Present study |
| Mean (±95% CI) | | | | | 61.1 (46.4–75.4) | | | | |
| With community management | Single season | – | 6 | – | 28.2 | 383 | 108 | Aguarico River, Ecuador | *Caputo, Canestrelli & Boitani (2005)* |
| | Multi season | 100 | 1 | 1 | 19.4 | 273 | 53 | Taboleiro da Água Preta, Brazil | *Pignati et al. (2013)* |
| | Multi season | – | 4 | – | 0.1 | 676 | 1 | Iténez & Paraguá, Bolivia | *Lipman (2008)* |
| | Single season | 42.3 | 26 | 11 | 26.4 | 144 | 38 | Falsino River, Brazil | Present study |
| Mean (±95% CI) | | | | | 18.5 (6.7–27.8) | | | | |

Notes.
[a]Means not calculated for area harvest rates due to small sample sizes. Dashes indicate when values were not reported.

significantly compared with the reference year (2011, 75%) with no enforcement and no community-based management (Table 1, 2-sample test for equality of proportions, $P < 0.0001$).

## DISCUSSION

Our findings strongly suggest that law enforcement patrols as a nest protection strategy have little effect on river turtle nest harvesting. Additionally, multiple lines of evidence suggest that there is no direct cause and effect relationship between enforcement effort and nest harvest rates. In contrast, community management was associated with a significant reduction in nest harvest rates.

A first year (egg and hatchling) survival rate of 0.2 is typical for population growth in river turtles (*Iverson, 1991*; *Pike et al., 2008*; *Zimmer-Shaffer, Briggler & Millspaugh, 2014*). Although we found that nest harvest rates did decline during years with enforcement compared with the 2011 reference level, nest harvest levels remained so high that they are likely to be unsustainable for populations even under the best-case scenario of no adult harvest. For example, harvest rates increased in the second year of enforcement to 67%. This implies that no more than an additional 13% of the original nest cohort could die as hatchlings before first year survival would fall below the 0.2 survival threshold and the population would enter a decline phase. Such a low hatchling mortality rate is unlikely considering the challenges for survival of this small and relatively fragile stage (*Iverson, 1991*) and the diversity of aquatic predators in Amazon rivers. Therefore our findings suggest that the governance of river turtle management plans with external enforcement was ineffective in the area studied.

The use of external enforcement has been widely applied and documented for the governance of forestry and fisheries management across Amazonia (*Lambin et al., 2014*; *McGrath et al., 2015*; *Nepstad et al., 2006*; *Nolte et al., 2013*; *Peres, Barlow & Laurance, 2006*; *Richards et al., 2017*). In Brazil such enforcement is also widespread (*McGrath et al., 2015*; *Richards et al., 2017*) and both the local traditional riverine communities and townspeople in our study area are familiar with the actions of external enforcement agents. Our findings strongly suggest that the enforcement mode as applied in our study area was not suitable to detect and/or deter illegal nest harvesting activities. The enforcement approach was typical for Amazon waterways and relied heavily on diurnally operated, motorized boat patrols. Female turtles generally lay eggs at night, and in our study area locals report that harvesters often wait at nesting areas overnight and collect nests almost immediately as eggs are laid. This means that harvesters are unlikely to be detected by the diurnal enforcement patrols. Additionally, as harvest rates increased in the second year it is also possible that the harvesters became more familiar with the enforcement patrols and more confident in their ability to avoid interdiction.

The clear patterns observed are unlikely to be artifacts of the sampling approach used. There is no evidence to suggest that differences observed in number of turtle nesting areas and nests were due to observer bias during the searches as we maintained at least one observer in the team constant while conducting searches in all years (2011, 2015, 2016, and 2017). Similarly, there is no evidence to suggest the magnitude of the differences between years can be explained by natural variation in the number of turtle females and therefore number of nests during the survey period. In fact, considering the well-documented anthropogenic impacts on the wildlife community in our study area (*Norris & Michalski, 2013*; *Norris, Michalski & Gibbs, 2018*) it is more reasonable to expect a decline in the overall number of nests.

Nest harvests by humans is ubiquitous across the species range (*Escalona & Fa, 1998*; *Hernández et al., 2010*; *Landeo, 1997*; *Smith, 1979*; *Vogt, 2008*); without management the levels of human harvest of river turtle nests are typically >50% (Table 2) and can reach 100% at nesting areas (*Bermúdez-Romero et al., 2010*; *Hernández et al., 2010*; *Lipman, 2008*) (D Norris, pers. obs., 2016). Although we observed high harvest levels prior to community

management, estimated nest harvest proportions are likely to represent minimum values. This is because nests sites can be hard to detect as they may be concealed post-harvest by harvesters. Therefore, differences in detectability might explain at least partially variation (harvest levels ranged from 55% to 75%) in the proportion of nests removed in the years prior to the community management in 2017.

Fewer nests were found during the years with enforcement patrols (2015 and 2016), a pattern we hypothesize can be attributed to the increased and/or more careful concealment of harvested nests as a response to the presence of the enforcement patrols. The harvest of nests falls within a poorly defined area of the protected area governance and legislation. The river is outside of the protected area border and Brazilian legislation allows for the harvest of natural resources to meet basic (nutritional) needs. Although both the existing governance regime and legislation is often ambiguous and unclear, local residents are within their rights to consume the river turtle nests. Why, then, would nest harvest concealment increase with the presence of enforcement? The most likely explanation is that nest harvest was not carried out by the local residents. Although interviews reveal that more than 50% of local residents eat turtle eggs, the locals remain close (typically < 500 m) to their houses (*Norris & Michalski, 2013*). Community members cite harvest for commercial exploitation by outsiders (town residents) as the main cause of nest removal.

The community management project that was implemented did not directly target human removal of turtle nests. So why then was there such a sharp reduction in nest harvest? It is important to note that the community management project was inspired by the community members after previous management and governance approaches did not provide outcomes desired either for species conservation or local aspirations for community development. The community had expressed concern regarding environmental degradation in the area including the loss of turtle nests (due to human harvest *Norris & Michalski, 2013* and the submersion of nesting areas by a newly installed hydroelectric dam *Norris, Michalski & Gibbs, 2018*), and the increasing amount of rubbish along the river at the beaches /nesting areas. Protecting the nests temporarily against natural predators was a way that community members could actively participate not only in caring for the turtle nests but also the surrounding environment. As nest harvesting by humans was not specifically targeted these actions had general support and conflicts were not generated. The lack of conflicts is also explained by the fact that community members do not depend on river turtle nests for their daily nutritional requirements or economic well-being (*Norris & Michalski, 2013*).

Providing payments for protecting nests and/or the selective harvest of nests that would otherwise be flooded have been used to engage local communities in the management of river turtle nests (*Caputo, Canestrelli & Boitani, 2005*). Different to such studies, we did not provide any financial rewards for participation, nor did we translocate nests for headstarting incubation nor sanction harvest of a subset of nests. Seasonal differences in our study area compared with that of *Caputo, Canestrelli & Boitani (2005)* partly explain the difference in approach. In our study area, peak nesting (mid to late October) takes place approximately two months before river levels rise (mid to late December), which means many turtle embryos are in advanced stages of development and the eggs are not

suitable for harvest when levels rise. This is because locals prefer fresh eggs, with harvesting activities also peaking around October. The uncertainty in future effects of climate change, changes in flow rates due to development patterns (e.g., hydropower developments *Timpe & Kaplan, 2017*), deforestation, and their synergistic effects on wildlife species and human populations are therefore a challenge for the implementation of conservation solutions. Such uncertainties reinforce the need for solutions to be tailored to the local context.

We discovered that the involvement of a relatively small number of key personnel had a broad impact and that a positive community perception (of doing the right thing) was sufficient to ensure engagement. Previous studies show that harvest and consumption of nests is not random within or between rivers (*Escalona & Fa, 1998*; *Hernández et al., 2010*; *Norris & Michalski, 2013*). Harvest rates are not spatially uniform, increasing at beaches closer to towns and in more accessible river sections (*Escalona & Fa, 1998*; *Hernández et al., 2010*; *Pignati et al., 2013*). Additionally, patterns of consumption are also aggregated within communities, with river turtle egg consumption by neighbors the strongest of 12 environmental, spatial and social variables used to explain patterns of river turtle egg consumption in the local community (*Norris & Michalski, 2013*). A detailed understanding of the local context and spatially explicit monitoring of nesting beaches and community activities is therefore required to ensure the success of any community-based management of river turtle nests.

Local communities living along Amazon rivers have increasing access to alternative food sources including poultry (mainly chicken) and farmed fish (*De Jesus Silva et al., 2017*; *Piperata et al., 2011*) and depend less on relatively limited seasonal supplies such as turtle eggs to meet their nutritional requirements (*De Jesus Silva et al., 2017*). Human nest predation and egg consumption has long been recognized as both a threat (reducing recruitment and population size) and opportunity (a valuable resource, which generates stakeholder involvement in conservation) for the conservation and management of *P. unifilis* populations (*Caputo, Canestrelli & Boitani, 2005*; *Mittermeier, 1978*; *Smith, 1979*). However, with riverine communities likely to become progressively less dependent on turtle eggs as a food source (*Piperata et al., 2011*), conservation activities need to be developed that do not rely simply on the preservation of nests for subsequent commercialization.

Our results suggest that indirect benefits and intrinsic values placed by local communities can be as important as economic gain for the development of successful conservation actions aimed at maintaining natural resources. We did not adopt an approach of payments for riverine people during the community-based management activities, and turtle nest harvest rates did decrease markedly when compared with years with enforcement patrols. For these reasons, we are confident to link the success of the community-based management to riverine perceptions of intrinsic value of preservation of the forest, rivers and the wildlife they support as shown in the meetings with landowners in our study.

The needs of different users can generate conflicts in common pool resources (*Ostrom, 2015*). Our findings from the first year of community-based management were overwhelmingly positive and the conflicts anticipated did not materialize. In the region studied, there appears within the local community to be a strong degree of respect for

natural resources and an understanding of environmental problems. The community-based management was implemented after seven years of research and has been developed with the local communities. Based on our findings we hypothesize that respectful and practical engagement along with good will are the most important drivers in explaining the success of the first year. There is obviously no guarantee that this will continue, and there is a need to continually engage and work with local communities within an adaptive management framework with the capacity to respond to socio-economic changes as well as new and unforeseen challenges.

## CONCLUSIONS

Although our findings come from the first year of community management the clear reduction in river turtle nest harvest illustrates that a focus on community involvement can generate immediate benefits for conservation within multiuse protected areas. Our findings suggest that the presence of community members monitoring and protecting against natural predators was sufficient to deter the harvest by outsiders without generating any obvious conflicts about river turtle conservation. As such, we conclude that the good will, mutual understanding, and collaborative development of conservation initiatives between local communities, researchers and conservationists are the vital/keystone components for the success of conservation activities within the sustainable-use protected areas.

## ACKNOWLEDGEMENTS

The Instituto Chico Mendes de Conservação da Biodiversidade (ICMBio) and the Federal University of Amapá (UNIFAP) provided logistical support. We thank the staff at ICMBio (Ivan Machado de Vasconcelos, Sueli Gomes Pontes) for their help in developing the community management activities and providing data on the enforcement effort during the study period. We are grateful to various volunteers, graduate and post-graduate students who have participated in data collection over the years. We also thank Alvino Pantoja Leal, Cremilson, Cleonaldo and Cledinaldo Alves Marques for their invaluable assistance during fieldwork.

### Funding

Funding was provided by the United States National Academy of Sciences and the United States Agency for International Development through the Partnership for Enhanced Engagement in Research (http://sites.nationalacademies.org/pga/peer/index.htm), and award number AID-OAA-A11-00012 to Darren Norris, James P. Gibbs and Fernanda Michalski. Fernanda Michalski received a productivity scholarship from CNPq (Process 301562/2015-6). The funders had no role in study design, data collection and analysis, decision to publish, or preparation of the manuscript.

## Grant Disclosures

The following grant information was disclosed by the authors:

United States National Academy of Sciences and the United States Agency: AID-OAA-A11-0001.

CNPq: Process 301562/2015-6.

## Competing Interests

The authors declare there are no competing interests.

## Author Contributions

- Darren Norris conceived and designed the experiments, performed the experiments, analyzed the data, prepared figures and/or tables, authored or reviewed drafts of the paper, approved the final draft.
- Fernanda Michalski conceived and designed the experiments, performed the experiments, prepared figures and/or tables, authored or reviewed drafts of the paper, approved the final draft.
- James P. Gibbs conceived and designed the experiments, performed the experiments, authored or reviewed drafts of the paper, approved the final draft.

## Field Study Permissions

The following information was supplied relating to field study approvals (i.e., approving body and any reference numbers):

Permission to collect observational data from river turtle nest-areas was provided by research permit number IBAMA/SISBIO 49632-1 and 49632-2 to DN and FM, issued by the Instituto Chico Mendes de Conservação da Biodiversidade (ICMBio). Interviews with local residents were approved by IBAMA/SISBIO (permits 45034-1, 45034-2, 45034-3) and the Ethics Committee in Research from the Federal University of Amapá (UNIFAP) (CAAE 42064815.5.0000.0003, Permit number 1.013.843).

## Data Availability

The raw data are provided in the Tables and Figures.

## Supplemental Information

Supplemental information for this article can be found online at http://dx.doi.org/10.7717/peerj.4856#supplemental-information.

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
