# Peer review of "Community involvement works where enforcement fails: conservation success through community-based management of Amazon river turtle nests"

_PeerJ, doi:10.7717/peerj.4856_

## Round 0.1 · original submission · Major Revisions

The manuscript is being returned for major revision. The reviewers have provided insights into problem areas and offer suggestions for revision.

For example, Reviewer 1 notes:

The research question is well defined, but the design of the study is not. 1- The authors only provide % of nests harvested and # of nests not harvested per year. They do not provide # of beaches surveyed, # of nest per beach per year, # of nests harvest per beach per year. Without this information, I cannot judge whether the work was rigorous or whether the finds are valid. What is the n of the study? What is the unit of measurement?

Reviewer 2 notes:

The authors fail to provide a proper description of their case. This may be because they are too familiar with the case themselves. However, this is not an excuse for not telling the reader the necessary details about e.g. the study area social setting, the community or communities involved in the management initiative, other relevant stakeholders, the roles of different actors in the management work, etc.

We look forward to the revised manuscript.

Reviewer 1 ·

Basic reporting

In general, the text is written in good English. However, there are some inconsistencies in spelling and punctuation. I have not tried to spot all of them not being a native English speaker myself. Please check the language again.

There are some inconsistencies in referencing. The authors should revise the references and make sure they are consistent throughout (e.g. chronological order of sources cited). The references themselves are well justified.

The article structure follows the PeerJ standard. Content-wise, the main weakness of the manuscript relates to how the argument is elaborated. The authors fail to provide a proper description of their case. This may be because they are too familiar with the case themselves. However, this is not an excuse for not telling the reader the necessary details about e.g. the study area social setting, the community or communities involved in the management initiative, other relevant stakeholders, the roles of different actors in the management work, etc. I have indicated these information needs in my detailed comments to the annotated manuscript pdf. Another major problem is that the readers fail to define well their key concepts. This is something I comment in more detail in the pdf, too. How are the key concepts defined and how they relate to each other in this particular study? To make the line of argument convincing and scientifically sound, the concepts and the background need to be revised.

The tables and figures are appropriate, but only if the above revisions are successfully made.

The authors provide no raw data. The editor may assess whether it is in line with the PeerJ policy in this case.

Because of the above reasons, in its current form, the manuscript is not entirely ‘self-contained,’. Its line of argument is hard to follow without knowing more about the case itself. However, I think it can be elaborated to stand alone and that it can represent an appropriate ‘unit of publication’. A major revision is needed.

Experimental design

The manuscript is a piece of original primary research. However, I am not sure how it differs from a ”case study” – something PeerJ is supposed not to publish. In my view, most studies in this field are case studies, with one or more cases involved. I assume that this type of study design nevertheless is considered to be within the aims and scope of the journal.

The paper suffers from weaknesses in the definition of its main research question(s). This mainly relates to the unclear conceptual basis and the insufficient background. Both in relation to the case itself and in relation to similar cases elsewhere. Doing this helps you to come up with well defined, relevant and meaningful research questions. This also helps you to answer how your research fills an identified knowledge gap.

The authors need to better describe their study methods and materials. This should take place along with conceptual and background revisions leading to better elaborated study questions. The field study permits as described by the authors seem appropriate to me.

Validity of the findings

The quality of the data cannot be easily evaluated (no raw data was provided). Better data (methods) is needed.

My detailed comments in the annotated pdf show my main concerns about how the conclusion are stated, linked to original research question and limited to supporting results.

Additional comments

I welcome the paper as an interesting piece of work on an important topic. In my view, in its current form the manuscript suffers from such major weaknesses that it requires a complete overhaul (major revision) for the editor to make a decision about its publishing in PeerJ. All of my comments and suggestions are found in the attached annotated pdf of the manuscript.

As a summary, I would suggest that the authors:

1) Revise and rewrite the conceptual basis of their paper. All key concepts need to be well defined and their relationships to the elaboration of the hypotheses and the argument should be well established.

2) Elaborate a proper background section. This can be dividedf, for instance, between the intro and the materials and the methods sections. The background should deliver key information about the study area setting, the relevant stakeholders and their roles, institutional setting, the situation of the turtles including main threats, the details of the community involvement in management, and so on. I have pointed out these needs in the annotated pdf.

3) Better define the research questions and elaborate the line of argument from this to the hypotheses, the methods, the results, discussion and the conclusions. This seems quite obvious, but I think that the authors may be “too” familiar with their case – they forget that the readers most likely are not. This affects not only the line of argument but also the general interest of the paper.

Annotated reviews are not available for download in order to protect the identity of reviewers who chose to remain anonymous.

Reviewer 2 ·

Basic reporting

The topic of this work is certainly very relevant for conservation of freshwater turtles. The comparison between enforcement and community management initiatives is highly appropriate and important, as enforcement is not sufficient (and often not effective) to control poaching of turtles and turtle eggs (and community initiatives for protection of beaches is one strategy used throughout the Amazon). That being said, I think the authors fail to provide sufficient details about the work for me to assess the quality and validity of the findings (see experimental design section).

Minor points:
Ln 49 – You do not mention all the efforts and suggestions on the demand side of the issue (e.g., target consumer behavior, increase alternative sources of food for consumers, decrease price of alternatives). You focused only on the supply side, which is not the only way to help conserve wildlife.
Ln 95 – You actually obtained ethical approval by an ethics committee. You always have to get that approval when you interview people, even if you don’t perform experiments with them. However, I missed where the data of the interviews are. You did not present these data.
Ln 165 – Are you taking the effort (liters of fuel) used for the entire region or for that portion you studied? It seems to me that if you are comparing only that portion, you should include the effort only for that portion. Is that what you did? If you used the effort for the entire region, please justify.

Experimental design

The research question is well defined, but the design of the study is not.
1- The authors only provide % of nests harvested and # of nests not harvested per year. They do not provide # of beaches surveyed, # of nest per beach per year, # of nests harvest per beach per year. Without this information, I cannot judge whether the work was rigorous or whether the finds are valid. What is the n of the study? What is the unit of measurement? The authors present % of nests harvested, but I would think that beach (or location surveyed) is the unit. If someone comes to harvest eggs on a beach, the harvest on that beach will be correlated. In this case, nest is not the sampling unit, but the beach might be (if beaches are not also spatially autocorrelated with regards to harvest – if they are, then n potentially equals 1).
2- The authors conducted sampling or census in the study area? I imagine they conducted a census as they did not use a statistical test to assess differences between the treatments. However, P. unifilis is known to nest in different habitats, not only on beaches. Did you survey all possible habitats (e.g. river banks?). Did you find all nesting habitats? I find it hard to believe that you would be able to map all nesting locations of this species, which can change from year to year. Please provide a description of your sampling/census. The authors should not expect me to read another paper to understand the basic information of sampling design they used.
3- The authors do not have a comparison/control group. I understand that sometimes it is not possible to have a comparison, but that only increases the need to be very clear about the design. What if number of nests varied widely (but naturally) from year to year? Fig 2 shows that 2011 and 2017 had higher number of nests, and the enforcement years had a small number of nests. What if the year of community management had a higher number of nests to begin with (compared to enforcement years)? You say that “Fewer nests were found during the years with enforcement patrols (2015 and 2016), and we hypothesize that these reductions can be attributed to the increased and/or more careful concealment of harvested nests as a response to the presence of the enforcement patrols” (Ln 242-244). But, if instead, the number of nests during enforcement years was naturally lower than the previous years (independent of treatment)? You don’t have a comparison to know that. Can you provide information that would address/ cast doubt on these potential confounding factors?

4- Ln 233-253 - Yes, people can conceal the harvest. With that in mind, if community members helped you to monitor harvest (Ln 152-154), then they knew exactly what you were looking for and were in the position, if they wanted to, to conceal the harvested nests better than anyone who harvested in other years. How do you know that this is not the case and that it is not why you had a lower rate of harvest? You are assuming people concealed the harvest more during enforcement years, but what if nest harvest was actually more frequently concealed during the year of community management? Then your results of % of nests harvested would be confounded with that.

Validity of the findings

1- I cannot assess the validity of the findings without more detailed information about the sampling design.

2 – The authors concluded that “good will, understanding, and perceptions of local communities are the vital/keystone components for the success of conservation activities” (Ln 268-270). Did you measure these variables? If so, please present the data. If not, I would suggest the authors limit these claims as just hypotheses.

---

## Round 0.2 · Minor Revisions

Thank you for your revised manuscript. It has come to our attention that not all of Reviewer 1's 58 comments in the annotated PDF were seen/addressed in the last rebuttal letter and revision. We are sending Reviewer 1's annotated PDF again with all comments attached to the highlighted portions. Please address all 58 of these comments. If you have any questions, please email [email protected].

---

## Round 0.3 · accepted · Accept

Congratulations! Your revised manuscript has been accepted for publication. Thank you for addressing reviewer comments that while extensive improved the manuscript. Preservation of Amazonian fauna is an important topic that receives far less attention than it deserves. Your manuscript demonstrates that community efforts can be successful in conservation efforts.

# Reviewer 1 ·

Basic reporting

No additional comments

Experimental design

No additional comments

Validity of the findings

No additional comments

Additional comments

I am happy to see that you have succesfully addressed (most of) my criticism. It was not my fault that you did not notice all of my comments in your first revision. Please understand that a proper response to all of the comments made by a reviewer is necessary -- as you know this is voluntary work and the time and dedication of the reviewer should be appreciated, even when the authors disagree on some of the points made. With the current revision (in practice the first round of changes), I think the article is much more universally interesting compared to the original version. In particular, this concerns the institutional aspects of the case. I have made seven (7) minor comments/requests in the pdf. I think you should be more careful in your use of the concept of "community" and very shortly state what your "community" is, so that the reader understands better the institutional and social logic of the interventions made. Please consider this -- it should be easy to tackle the issue with a few well chosen words added. I think this is an interesting paper and I hope to see it published soon.

Annotated reviews are not available for download in order to protect the identity of reviewers who chose to remain anonymous.

Reviewer 2 ·

Basic reporting

This paper has relevant information about turtle conservation and community-based conservation that deserves publication.

I am mostly satisfied with the responses to the previous concerns raised. However, I still have a major concern regarding detection that I think should be dealt with or at least acknowledged as a potential source of bias. I explain this in more details in the experimental design section.

Ln 323-329: Although subsistence consumption of wildlife is allowed, many people are afraid of enforcement because the law is ambiguous and because of the negative history of local communities with enforcement agencies. Local communities often cite outsiders as the problem, but that does not mean outsiders are the main problem. Without further evidence of "outsiders" being the problem, I don't think the authors can make that claim. It is just a speculation and it should be stated as such.
Ln 379-385: I did not see any data on community values. Do you have these data? You mention meetings. Did you collect data during these meetings? If so, please add. If you are basing this only on your perceptions, please make that clear.
In lines 391-393 you say you are hypothesizing that "respectful and practical engagement along with good will are the most important drivers in explaining the success". Yes, this is a hypothesis. However, in lines 402-405 you say that "that the good will, mutual understanding, and collaborative development of conservation initiatives between local communities, researchers and conservationists are the vital/keystone components for the success of conservation". Do you have any data on the interviews/meetings that you could provide (quantitative or qualitative) to substantiate this claim? You say you performed meetings and interviews with community members, right? If you could add some information about these meetings and these interviews, the claims you are making about why the community-based effort is working would be substantiated. Otherwise, you are just basing these claims on your perceptions of what is happening (and that should be made clear).

Experimental design

Research question and methods are well described.

Regarding detection probability, I understand that the authors used a consistent method and used at least one person throughout the study to decrease bias. However, detection probability does not depend only on the consistency of methods or observer bias, it depends highly on environmental conditions. For instance, the probability of detection can vary greatly from year to year depending on how dry or wet the areas were, on how much vegetation there was, on the level of the river, etc. Given that you cannot possibly detect all nests (which the authors themselves acknowledged previously), they cannot assume detection probability remained the same across the years.

If your data allows you to measure detection probability, then you can eliminate that issue. If not, you need to explain how the potential changes in detection probability (due to environmental conditions, not due to observer bias or changes in methods) may or may not affect the results.

Validity of the findings

If the authors can adequately address the issue of detection, the results can be considered robust and valid.